# Potential window alignment regulating ion transfer in faradaic junctions for efficient photoelectrocatalysis

Hongzheng Dong[1,5], Xiangyu Pan[2,5], Yuancai Gong [2], Mengfan Xue[3], Pin Wang[3], SocMan Ho-Kimura[4], Yingfang Yao [1], Hao Xin [2] ✉, Wenjun Luo [1] ✉ & Zhigang Zou [1,3]

In the past decades, a band alignment theory has become a basis for designing different high-performance semiconductor devices, such as photocatalysis, photoelectrocatalysis, photoelectrostorage and third-generation photovoltaics. Recently, a faradaic junction model (coupled electron and ion transfer) has been proposed to explain charge transfer phenomena in these semiconductor heterojunctions. However, the classic band alignment theory cannot explain coupled electron and ion transfer processes because it only regulates electron transfer. Therefore, it is very significant to explore a suitable design concept for regulating coupled electron and ion transfer in order to improve the performance of semiconductor heterojunctions. Herein, we propose a potential window alignment theory for regulating ion transfer and remarkably improving the photoelectrocatalytic performance of a MoS$_2$/Cd-Cu$_2$ZnSnS$_4$ heterojunction photocathode. Moreover, we find that a faradaic potential window, rather than the band position of the intermediate layer, is a criterion for identifying interface charge transfer direction. This finding can offer different perspectives for designing high-performance semiconductor heterojunctions with suitable potential windows for solar energy conversion and storage.

In the past several decades, photocatalysis, photoelectrocatalysis, photoelectrostorage, third-generation photovoltaics (such as quantum-dot or dye-sensitized, organic and perovskite solar cells) have attracted wide interest due to their low cost and high efficiency for converting solar energy into chemical or electric energy[1–5]. Among them, different kinds of heterojunctions, such as type-II heterojunction[6] and Z-scheme junction[7], have been widely used to separate photo-excited carriers in light absorbers (Supplementary Fig. 1). In all of these previous studies[6–8], a band alignment theory is a basis for manipulating interface electron transfer in high-performance devices. However, Luo et al. have found that faradaic junction interface charge transfer process (coupled electron and ion transfer) widely exists in these solar energy conversion and storage devices[9–17]. Since the classic band alignment theory only regulates electron transfer, it is not suitable for explaining coupled electron and ion transfer in faradaic junctions. Therefore, it is very significant to explore a suitable

[1]Eco-materials and Renewable Energy Research Center (ERERC), National Laboratory of Solid State Microstructures, College of Engineering and Applied Sciences, Nanjing University, Nanjing 210093, China. [2]State Key Laboratory for Organic Electronics and Information Displays, College of Chemistry and Life Sciences, Nanjing University of Posts & Telecommunications, Nanjing 210023, China. [3]Eco-materials and Renewable Energy Research Center (ERERC), Jiangsu Key Laboratory for Nano Technology, National Laboratory of Solid State Microstructures and Department of Physics, Nanjing University, Nanjing 210093, China. [4]Institute of Applied Physics and Materials Engineering, University of Macau, Macau SAR, China. [5]These authors contributed equally: Hongzheng Dong, Xiangyu Pan. ✉e-mail: iamhxin@njupt.edu.cn; wjluo@nju.edu.cn

design concept for regulating coupled electron and ion transfer in order to improve the performance of the semiconductor heterojunctions.

In this work, we have improved the photoelectrochemical hydrogen production performance of a MoS₂/Cd-Cu₂ZnSnS₄(CCZTS) heterojunction photocathode significantly by introducing an Au intermediate layer. However, a Pd intermediate layer with a similar Fermi level does not improve the performance of the same MoS₂/ CCZTS heterojunction photocathode. The results suggest that the classic band alignment theory is not suitable for explaining the different effects of the Pd and Au layers on the photoelectrochemical properties of CCZTS. In order to investigate the reasons for the improved performance of MoS₂/Au/CCZTS heterojunction photo-cathode, some characterization methods, including in situ irradiated X-ray photoelectron spectra (XPS), quasi in situ irradiated Raman, in situ electrochemical Raman and time-of-flight secondary-ion mass spectrometry (TOF-SIMS), were carried out. We find that the MoS₂ plays the role of recombination centers by a revisable faradaic reaction and decreases the performance of a MoS₂/CCZTS heterojunction photocathode. The introduction of the Au intermediate layer inhibits alkali metal ion intercalation reaction of MoS₂ and decreases the recombination in the MoS₂/CCZTS heterojunction, but the Pd inter-mediate layer does not. Further studies suggest the Au layer indicates negligible reduction faradaic potential window, compared to the Pd layer, which suppresses alkali metal ion intercalation into MoS₂. Therefore, the theory of the potential window alignment can be used to explain the interface charge transfer in photoelectrocatalysis and can also provide guidance to improve the performance of photo-catalysis, photoelectrostorage, third-generation photovoltaics and so on.

## Results

### The Au intermediate layer to improve photoelectrocatalytic performance of a MoS₂/CCZTS heterojunction photocathode remarkably

In previous studies, Mo glass was widely used as a substrate for the preparation of a CCZTS thin film due to its good mechanical strength at high temperature[18,19]. Here, CCZTS films were deposited on Mo glass substrates by spin-coating and post-sulfurization method. After the sulfurization at high temperature, MoS₂ was inevitably in situ pro-duced on the surface of Mo glass substrates, which contacts with CCZTS to form MoS₂/CCZTS heterojunction[20,21]. For MoS₂/Pd/CCZTS and MoS₂/Au/CCZTS samples, Pd and Au intermediate layers were deposited firstly on Mo glass substrates with an ion beam sputtering method and then CCZTS layers were prepared. The preparation details of the three samples can be found in the Methods section. The three samples were characterized by XRD (Supplementary Fig. 2a). The dif-fraction peaks at 28.4°, 47.3° and 56.2° correspond to the (112), (220) and (312) planes of kesterite CCZTS[22,23], and the peaks of Au can also be observed in MoS₂/Au/CCZTS sample. However, no diffraction peaks of Pd and MoS₂ are observed in the three samples by the conventional XRD method due to their poor crystallinity. In order to further identify the Pd and MoS₂, the CCZTS layers were removed by a tweezer and Grazing Incidence X-Ray Diffraction (GIXRD) was used to characterize the exposed MoS₂ and MoS₂/Pd substrates (Supplementary Fig. 2b). By this method, the diffraction peaks of Pd and 2H-MoS₂ are observed. Figure 1a–c show the cross-sectional SEM images of MoS₂/CCZTS, MoS₂/Pd/CCZTS and MoS₂/Au/CCZTS, respectively. In the three sam-ples, well-grown CCZTS grains with different sizes are packed on the substrates. Therefore, the Pd and Au intermediate layers do not change the morphologies of CCZTS obviously. Moreover, both Pd and

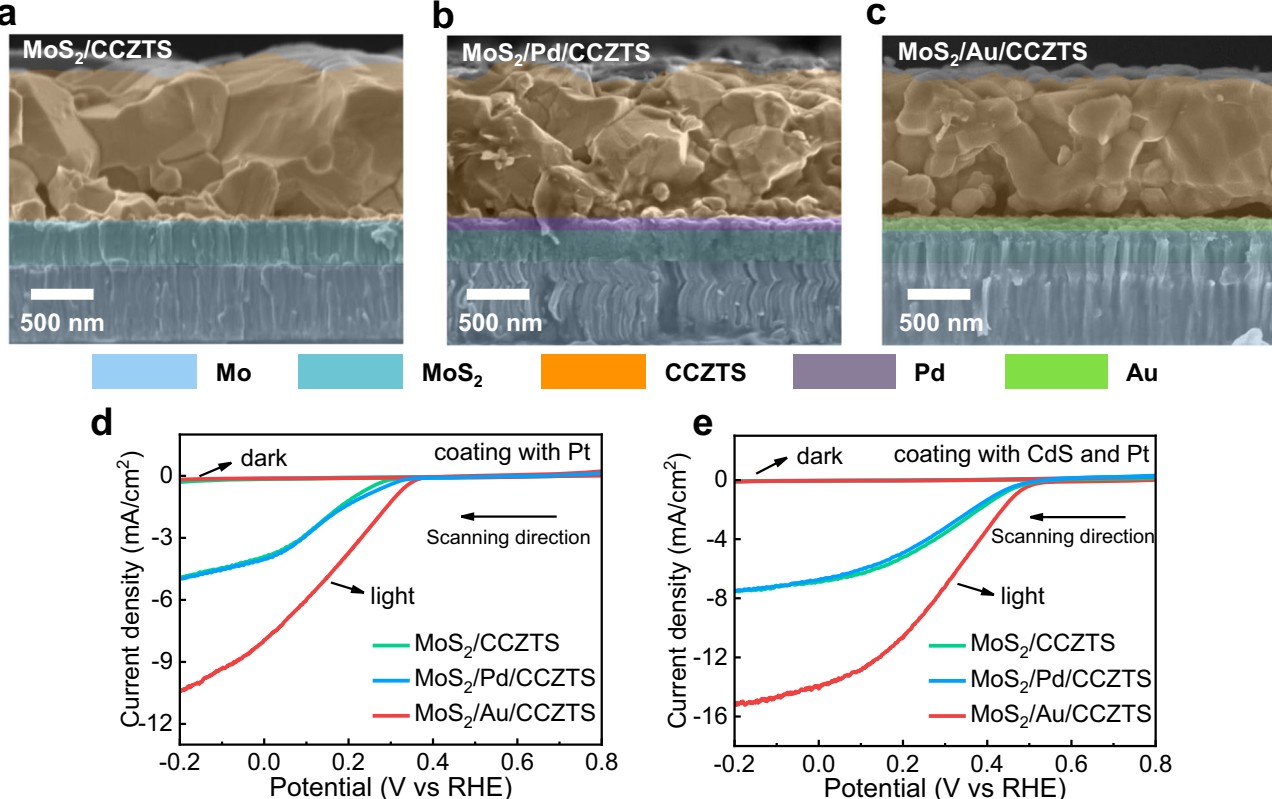

**Fig. 1 | Characterization and photocurrent of MoS₂/CCZTS, MoS₂/Pd/CCZTS and MoS₂/Au/CCZTS samples. a–c** The cross-sectional SEM images of MoS₂/ CCZTS (**a**), MoS₂/Pd/CCZTS (**b**) and MoS₂/Au/CCZTS (**c**). **d**, **e** Linear sweep vol-tammetry curves of MoS₂/CCZTS, MoS₂/Pd/CCZTS and MoS₂/Au/CCZTS after coating a Pt co-catalyst (**d**) or coating a CdS buffer layer and a Pt co-catalyst (**e**); Scan rate of 10 mV/s; Electrolyte: 0.2 M Na₂HPO₄/NaH₂PO₄ aqueous solution (pH = 6.5). Light source: an AM 1.5 G sunlight simulator, light intensity: 100 mW/cm².

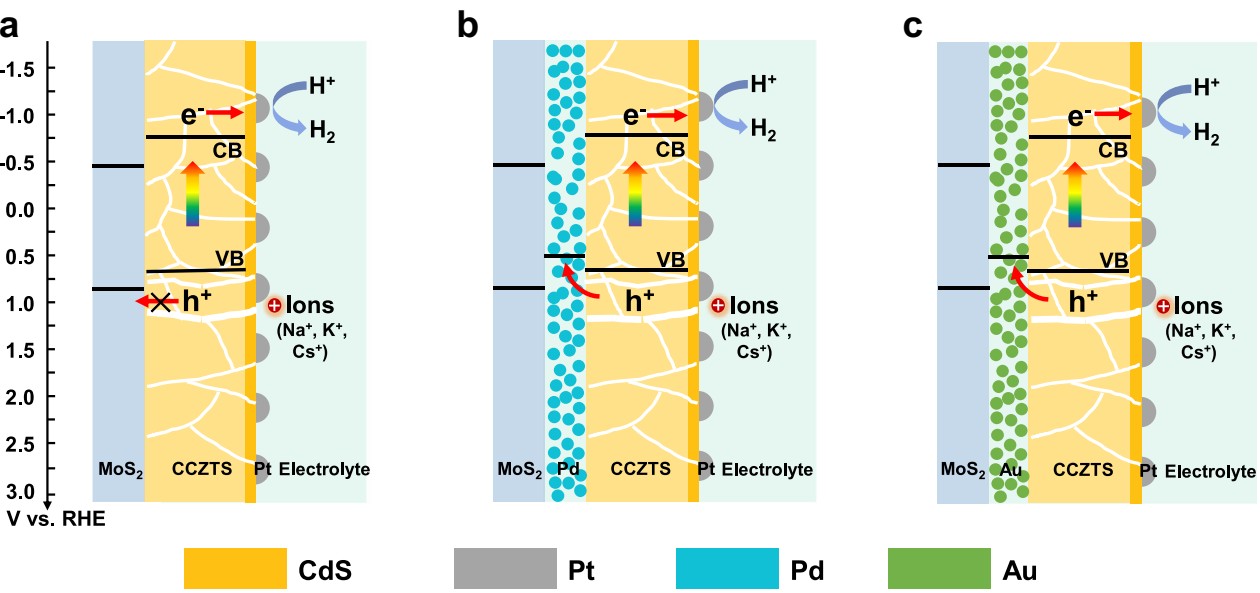

**Fig. 2 | Conventional band alignment theory for interface charge transfer. a–c** The diagram of interface charge transfer in MoS$_2$/CCZTS (**a**), MoS$_2$/Pd/CCZTS (**b**) and MoS$_2$/Au/CCZTS (**c**) heterojunction photocathodes.

Au intermediate layers between MoS$_2$ and CCZTS indicate small particle aggregates, which are about 80 nm in thickness (Supplementary Figs. 3, 4) and cover the MoS$_2$ underlayers incompletely in the heterojunctions from the top-view SEM images by removing the upper layers of CCZTS (Supplementary Fig. 5).

In order to investigate the effects of Pd and Au intermediate layers on the photoelectrochemical performance of CCZTS-based photocathodes, the photocurrent density of MoS$_2$/CCZTS, MoS$_2$/Pd/CCZTS and MoS$_2$/Au/CCZTS samples were measured in 0.2 M Na$_2$HPO$_4$/NaH$_2$PO$_4$ aqueous solution. From previous studies[24], the photocurrent of the bare CCZTS sample is negligible due to its low electrocatalytic activity for hydrogen production. Therefore, Pt co-catalysts were loaded on the surfaces of the three samples by an ion beam sputtering method for photoelectrocatalytic measurement. Figure 1d indicates Linear sweep voltammetry (LSV) curves of the three samples with Pt cocatalysts. Usually, a more negative potential has a larger driving force for electron-hole separation in a photocathode, which will lead to a higher photocurrent at a more negative potential. Therefore, the photocurrent of all of the three samples increases with shifting the potential negatively. It is worth noting that the photocurrent of CCZTS increases significantly after introducing the Au layer. In contrast, the MoS$_2$/Pd/CCZTS exhibits a similar photocurrent to that of the MoS$_2$/CCZTS photocathode. Moreover, n-CdS was also introduced as a buffer layer between CCZTS and Pt co-catalysts to further improve the photoelectrocatalytic performance[22,23]. The photocurrent density of MoS$_2$/CCZTS increases from 7.2 mA/cm$^2$ to 14.9 mA/cm$^2$ @ 0 V$_{RHE}$ after introducing the Au intermediate layer, but slightly decreases after introducing the Pd layer (Fig. 1e), which is further confirmed by the statistical results in the Supplementary Fig. 6. Moreover, the three samples also indicate good stability under illumination and the photocurrents only decay by about 5–11% after 3600 s of illumination (Supplementary Fig. 7). Therefore, the Au intermediate layer can improve the photoelectrocatalytic performance of the MoS$_2$/CCZTS heterojunction photocathode remarkably.

In order to investigate the reason that enhanced photocurrent is only observed in MoS$_2$/Au/CCZTS heterojunction photocathode, but not in MoS$_2$/Pd/CCZTS, we measured the substrate resistance and light absorption of the samples and the results are shown in Supplementary Table 1 and Supplementary Fig. 8, respectively. The MoS$_2$/Au substrate exhibits similar resistance as the MoS$_2$/Pd substrate and the light

absorption of CCZTS does not change obviously on different substrates, suggesting that the difference in performance of these samples does not come from the change of the substrate resistance and light absorption. Since a band alignment theory is usually used to understand the improved performance in the heterojunction, the band gaps and band positions of four CCZTS and MoS$_2$ samples were measured by UV-vis spectra and Mott-Schottky methods, respectively (Supplementary Figs. 9, 10). The average values and errors of the band gaps and band positions of the four samples are calculated and shown in Supplementary Table 2, which are in good agreement with some previous reports[23,25]. Moreover, the Fermi levels (E$_F$) of Pd and Au particulate films in this study were obtained by ultraviolet photoelectron spectroscopy (Supplementary Fig. 11) and the values are summarized in Supplementary Table 3, which are similar to the reference values in previous studies[26,27]. According to the above values, the band positions of MoS$_2$/CCZTS, MoS$_2$/Pd/CCZTS and MoS$_2$/Au/CCZTS heterojunction photocathodes are plotted in Fig. 2. Based on the band alignment theory, the valence band of CCZTS is higher than that of MoS$_2$, unfavorable for photo-generated hole transferring from the CCZTS photocathode to MoS$_2$. Therefore, the Pd or Au intermediate layers with the higher Fermi level will be helpful for hole transfer from CCZTS to the metal layers, which can improve the performance of a CCZTS photocathode[28]. However, in experiments, not the Pd but the Au intermediate layer improves the photocurrent for solar hydrogen production. Therefore, the classic band alignment theory is not suitable for explaining the converse effects of the Pd and Au layers on the photoelectrochemical properties of CCZTS.

## The Au intermediate layer to regulate ion transfer in the MoS$_2$/CCZTS heterojunction photocathode under illumination

According to previous studies[9–16], a faradaic junction charge transfer widely exists in different heterojunctions. Herein, we investigated the effects of Pd and Au on interface charge transfer in MoS$_2$/CCZTS by in situ irradiated XPS and the experimental details are shown in Supplementary Fig. 12. Since XPS is a surface characterization technique, it is difficult to obtain the Mo signal in the substrate if the CCZTS upper layer is too thick and compact. A thinner and porous CCZTS upper layer was prepared by only reducing the number of coating layers from seven to two with the same method, and the XPS peak of Mo 3d becomes more intensive in the MoS$_2$/CCZTS sample (Supplementary

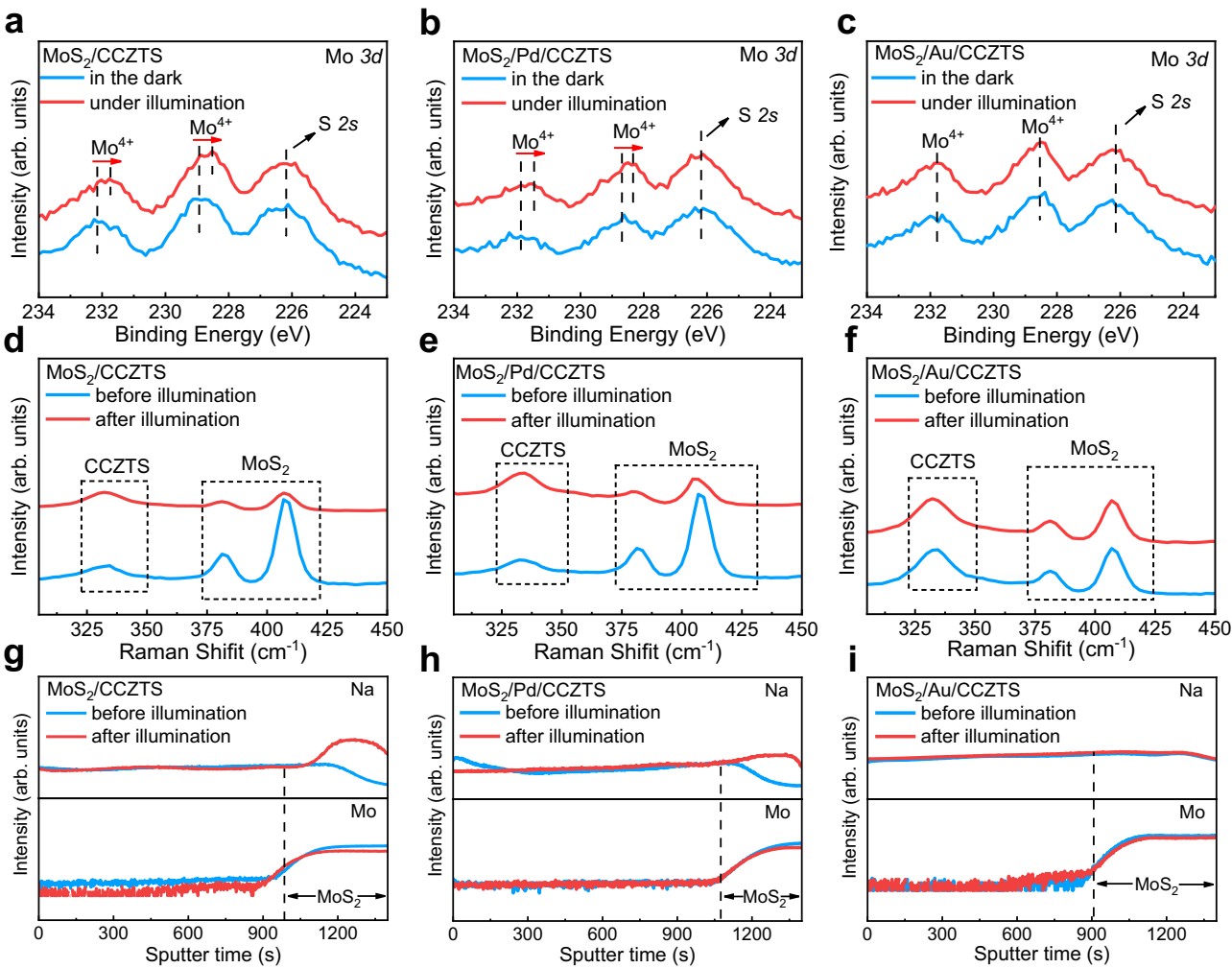

**Fig. 3 | Interface charge transfer process in MoS₂/CCZTS, MoS₂/Pd/CCZTS and MoS₂/Au/CCZTS samples. a–c** In situ irradiated XPS spectra of Mo *3d* in MoS₂/CCZTS (**a**), MoS₂/Pd/CCZTS (**b**) and MoS₂/Au/CCZTS (**c**) in the dark and under full arc Xe lamp illumination. The coating and annealing were repeated two times to obtain a detectable XPS signal of Mo. **d–f** Quasi in situ irradiated Raman spectra of MoS₂/CCZTS (**d**), MoS₂/Pd/CCZTS (**e**) and MoS₂/Au/CCZTS (**f**) in the electrolyte before and after full arc Xe lamp illumination. **g–i** Secondary ion intensities of Na and Mo ions in MoS₂/CCZTS (**g**), MoS₂/Pd/CCZTS (**h**) and MoS₂/Au/CCZTS (**i**) junctions in the electrolyte before and after full arc Xe lamp illumination. Electrolyte: 0.2 M Na₂HPO₄/NaH₂PO₄ aqueous solution; Light source: a full arc Xe lamp.

Fig. 13). Therefore, the MoS₂/CCZTS sample with two coating layers was investigated by in situ irradiated XPS and the results are shown in Fig. 3a–c. Two binding energies at 231.9 eV and 228.8 eV are assigned as $Mo^{4+}$ in the MoS₂/CCZTS sample in the dark (Fig. 3a), which shift to lower values under illumination[29]. The results suggest that $Mo^{4+}$ ions are photo-reduced into lower valence states[30]. After introducing Pd and Au intermediate layers, lower binding energies of Mo are only observed in MoS₂/Pd/CCZTS, but not in MoS₂/Au/CCZTS under illumination (Fig. 3b, c). The results suggest that the Au rather than Pd intermediate layer can prevent the reduction of $Mo^{4+}$ in the MoS₂/CCZTS junction. On the other hand, the binding energies of Cu *2p*, Zn *2p*, Sn *3d* and S *2p* in the three samples do not shift under illumination, suggesting that the CCZTS does not change (Supplementary Figs. 14–16). Quasi in situ irradiated Raman spectra were also used to investigate the effects of Au and Pd on the reduction process of MoS₂ in MoS₂/CCZTS before and after illumination[31] and the experimental details are shown in Supplementary Fig. 17. The Raman peaks at 382 cm⁻¹ and 408 cm⁻¹ are assigned as $E^1_{2g}$ and $A_{1g}$ vibrational modes of MoS₂, and the peak at 332 cm⁻¹ corresponds to CCZTS[22,29]. After MoS₂/CCZTS is illuminated, the Raman peak intensity of CCZTS does not change, while the Raman peak intensity of MoS₂ decreases (Fig. 3d). In contrast, the Raman peak intensity of MoS₂ film alone does not

decrease after the same illumination (Supplementary Fig. 18). Therefore, interface charge transfer in the MoS₂/CCZTS heterojunction under illumination leads to the change of the Raman peak intensity of MoS₂. According to previous studies[32,33], the decrease of Raman peak intensity of MoS₂ can be attributed to the Na⁺ intercalation into MoS₂. The Raman peak intensity of MoS₂ decreases in MoS₂/Pd/CCZTS but not in MoS₂/Au/CCZTS after illumination (Fig. 3e, f). The results further suggest that the Au rather than Pd intermediate layer can inhibit the reduction intercalation reaction of MoS₂ in the MoS₂/CCZTS junction, which are in good agreement with in situ irradiated XPS results mentioned above.

In order to further investigate the effects of Au and Pd on interface ion transfer in MoS₂/CCZTS under illumination, time-of-flight secondary ion mass spectrometry (TOF-SIMS) depth profiles were used to identify the element distribution in the three samples in 0.2 M Na₂HPO₄/NaH₂PO₄ aqueous solution before and after illumination. Before illumination, Na⁺ ions uniformly distribute in the bulk of the CCZTS film and decrease at the MoS₂/CCZTS interface. After illumination, the distribution of Cu, Zn, Sn and S in CCZTS does not change obviously (Supplementary Fig. 19), while the intensity of Na⁺ ions enhances remarkably at the depth of sputtering time >980 s in the sample (Fig. 3g), where the MoS₂ is located. Since some pin holes

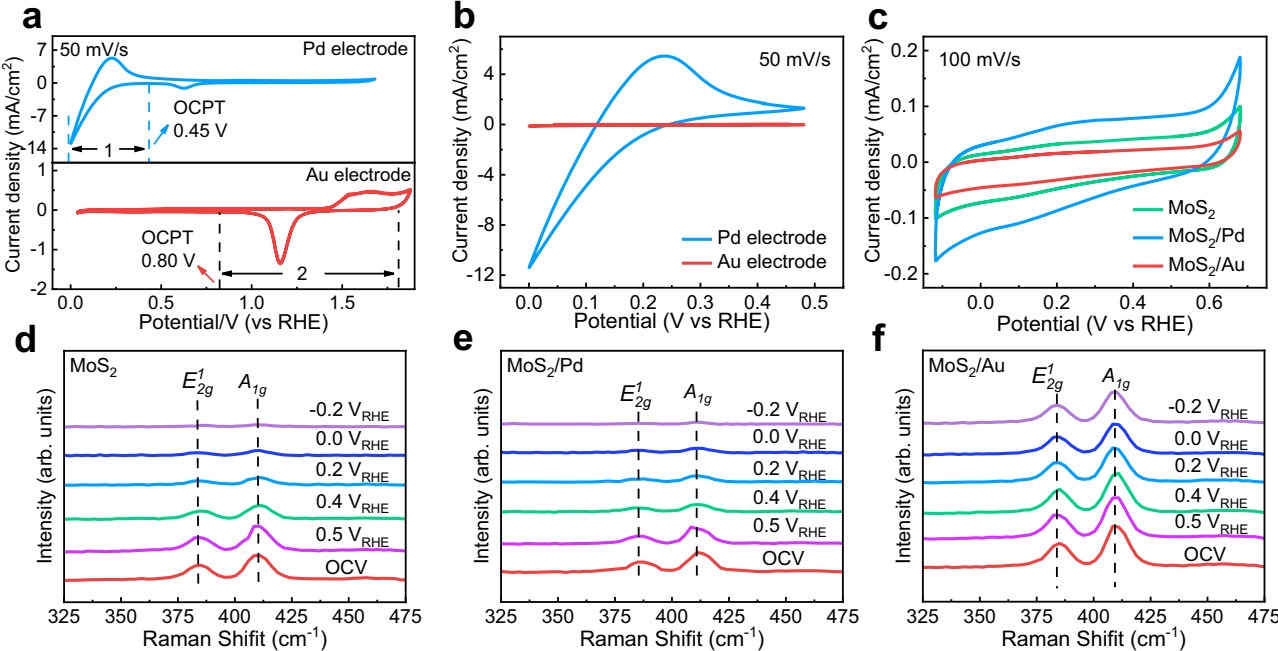

**Fig. 4 | The reasons of Au rather than Pd to inhibit Na⁺ intercalation into MoS₂.**
**a**–**c** Cyclic voltammetry curves of Pd and Au electrodes (**a**, **b**), MoS₂, MoS₂/Pd and MoS₂/Au (**c**) in the electrolyte in the dark. **d**–**f** In situ electrochemical Raman spectra of MoS₂ (**d**), MoS₂/Pd (**e**) and MoS₂/Au (**f**) in the electrolyte at different potentials in the dark. Electrolyte: 0.2 M Na₂HPO₄/NaH₂PO₄ aqueous solution.

inevitably exist even in the dense and thick CCZTS films prepared by the spin-coating method, electrolyte can penetrate into the MoS₂ layer through the pin holes (Supplementary Fig. 20). Therefore, the Na⁺ intercalation into MoS₂ happens in the MoS₂/CCZTS sample in the electrolyte under illumination. According to in situ irradiated XPS and TOF-SIMS, the MoS₂/CCZTS is a faradaic junction with the interface reaction of $MoS_2 + x\ Na^+ + x\ e^- \leftrightarrow Na_xMoS_2$. In the MoS₂/Pd/CCZTS junction, similar Na⁺ intercalation into MoS₂ was observed after illumination (Fig. 3h and Supplementary Fig. 21). In contrast, the distribution of Na⁺ in the MoS₂/Au/CCZTS junction does not change after illumination (Fig. 3i and Supplementary Fig. 22). Therefore, the Na⁺ intercalation into MoS₂ is suppressed only by the Au rather than Pd intermediate layer in MoS₂/CCZTS.

In order to further understand why the Au layer, not the Pd layer, can inhibit the faradaic reaction of MoS₂, the electrochemical behavior of Pd and Au were measured and the results are shown in Fig. 4a. According to previous studies[34,35], the faradaic potential window is defined as a potential range in which a material exhibits stable and reversible electrochemical faradaic redox reactions in the electrolyte. Therefore, a Pd electrode indicates a reduction faradaic potential window[16,36,37], which is assigned as $Pd + x\ H^+ + x\ e^- \leftrightarrow \beta\text{-}PdH_x$ (1). Different from a Pd electrode, an Au electrode only exhibits an oxidation faradaic potential window, $Au + H_2O \leftrightarrow Au\text{-}O + 2\ H^+ + 2\ e^-$ (2). Since MoS₂ is reduced by photo-generated electrons in CCZTS, enlarged reduction faradaic potential windows of Pd and Au are shown in Fig. 4b. A Pd electrode indicates a reduction faradaic potential window of 0 V_RHE to 0.45 V_RHE, while the Au electrode indicates a negligible reduction faradaic capacitance at the same potential window. It is because the lower adsorption energy of H⁺ on the surface of Pd than on the Au layer, which leads to a much higher capacitance of Pd than Au[38,39]. In order to investigate the effects of the coating of Pd and Au on the faradaic capacitance of MoS₂ (Supplementary Fig. 24), the cyclic voltammetry curves of MoS₂, MoS₂/Pd and MoS₂/Au were measured in the same electrolyte and the results are shown in Fig. 4c. The faradaic capacitance of MoS₂ increases after the coating of a Pd layer, but decreases obviously after coating the Au layer. In order to further elucidate the reasons of Au not Pd inhibiting

Na⁺ intercalation, in situ electrochemical Raman spectroscopy was carried out to investigate the change of MoS₂, MoS₂/Pd and MoS₂/Au samples during electrochemical measurement and the experimental details are shown in Supplementary Fig. 25. The Raman peak intensity of MoS₂ does not change obviously at the potential of 0.5 V_RHE, and decreases from 0.4 V_RHE to negligible at −0.2 V_RHE (Fig. 4d). Therefore, Na⁺ intercalation in MoS₂ becomes more intense with the potential shifting negatively, consistent with the previous study[33]. The Raman peak intensity of MoS₂ in a MoS₂/Pd sample also decreases with the potential shifting negatively, but changes only slightly in MoS₂/Au (Fig. 4e, f). In a MoS₂/Pd sample, after the reduction adsorption of H⁺ on Pd ($Pd + x\ H^+ + x\ e^- \leftrightarrow \beta\text{-}PdH_x$), interface charges can transfer from Pd to MoS₂ following the reaction of $MoS_2 + x\ Na^+ + \beta\text{-}PdH_x \leftrightarrow Na_xMoS_2 + Pd + x\ H^+$. In contrast, there is negligible reduction adsorption of H⁺ on an Au layer, which leads to Na⁺ intercalation being suppressed into MoS₂. Therefore, the Au layer plays the role of an ion blocking layer in the heterojunction photocathode. In order to further confirm the effects of an ion blocking layer, the photoelectrocatalytic performances of the MoS₂/CCZTS, MoS₂/Pd/CCZTS and MoS₂/Au/CCZTS photocathodes were also measured in the electrolytes with different alkali metal ions (Na⁺/K⁺/Cs⁺) (Supplementary Figs. 26–29). The results suggest that the intercalation reaction of MoS₂ can also be suppressed with increasing alkali metal ion radii in the electrolytes and improve the photoelectrocatalytic performance of the three heterojunction photocathodes. Especially, the MoS₂/Au/CCZTS photocathode indicates a high photocurrent of 16.3 mA/cm² at 0 V_RHE in the electrolyte with Cs⁺, which is close to the highest value in the neutral aqueous solution (Supplementary Table 4). Therefore, the photoelectrocatalytic performance of the heterojunction photocathodes can be improved remarkably by regulating ion transfer. So far, there are some reports on improving the performance of photoelectrocatalytic cells and dye-sensitized solar cells by adjusting the concentrations and species of ions in the electrolytes[40,41]. However, most of these previous studies only focus on the effects of various electrolytes on the properties of an individual catalyst or semiconductor, but not interface charge transfer in the heterojunctions. This work provides a different paradigm of

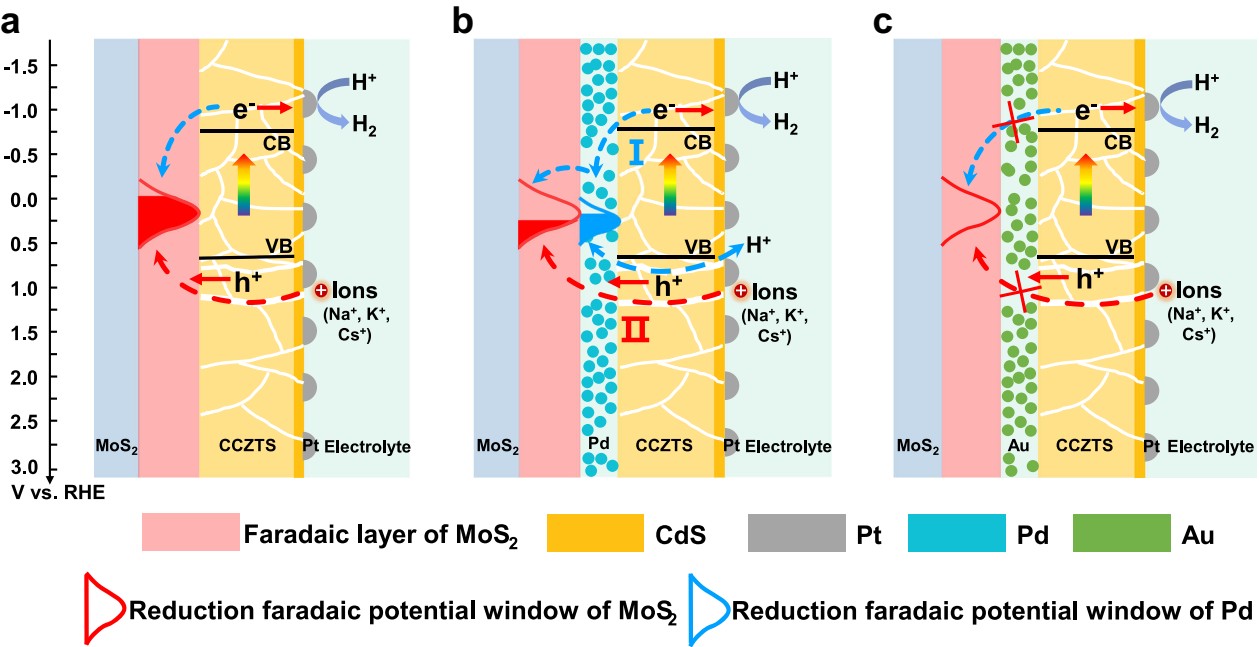

**Fig. 5 | The mechanism of Au to improve the photoelectrocatalytic performance of a MoS₂/CCZTS heterojunction photocathode. a–c** Interface charge transfer in MoS₂/CCZTS (**a**), MoS₂/Pd/CCZTS (**b**) and MoS₂/Au/CCZTS (**c**) heterojunction photocathodes after coating a CdS buffer layer and a Pt co-catalyst.

regulating ion transfer to design high-performance semiconductor heterojunctions.

## Potential window alignment to improve photoelectrocatalytic performance in a MoS₂/Au/CCZTS heterojunction photocathode

According to the above results and analysis, we propose a mechanism for the improved performance of MoS₂/CCZTS after introducing the Au intermediate layer. According to the above results (Supplementary Figs. 9, 10, 24), the band positions of semiconductors and the faradaic potential windows of MoS₂, Pd and Au are plotted in Fig. 5. When the conduction band of CCZTS is higher than the faradaic potential window of MoS₂, part of the photo-generated electrons in CCZTS can reduce H⁺ into hydrogen, and the other electrons reduce the MoS₂ with the intercalation of Na⁺ (K⁺/Cs⁺) from the electrolyte. Moreover, part of photo-excited holes can re-oxidize the $Na_xMoS_2$ ($K_xMoS_2$ or $Cs_xMoS_2$) during the transfer process. Therefore, the MoS₂ plays the role of recombination centers, which decreases the photocurrent of the MoS₂/CCZTS sample (Fig. 5a). Since a Pd intermediate layer indicates a similar reduction faradaic window position with MoS₂, the photo-generated electrons in CCZTS can transfer to Pd and induce the H⁺ adsorption reaction to form $\beta$-$PdH_x$, which can further reduce MoS₂ and is recovered into Pd. The charge transfer at MoS₂/Pd interface follows the reaction of $MoS_2 + x\ Na^+\ (K^+/Cs^+) + \beta\text{-}PdH_x \leftrightarrow Na_xMoS_2$ ($K_xMoS_2/Cs_xMoS_2$) + Pd + x H⁺, which leads to a higher capacitance and a lower photocurrent in MoS₂/Pd/CCZTS. However, the Au intermediate layer can inhibit the Na⁺ intercalation into MoS₂ and improve the performance of the MoS₂/CCZTS sample. These results suggest that a faradaic potential window, not a Fermi level, is a criterion for interface charge transfer direction in a faradaic junction. The driving force for charge separation in a faradaic junction is the difference of electrochemical potential at the interface, which is intrinsically different from the built-in electric field in a physical heterojunction based on classic band theory[9–16].

## Discussion

In summary, by introducing the Pd and Au intermediate layers in MoS₂/CCZTS, the photocurrent increases obviously in MoS₂/Au/CCZTS, but

decreases slightly in MoS₂/Pd/CCZTS. It is because the Au layer acts as an ion blocking layer to inhibit the recombination of photo-generated carriers and improve the photocurrent. Moreover, we find that it is the potential windows not the band positions of the intermediate layer that is the criterion for controlling interface charge transfer direction in a faradaic junction. The potential window alignment theory cannot only deepen the understanding of interface charge transfer in the heterojunction but also offer a suitable design concept for improving the performance of other solar energy conversion and storage devices, such as photocatalysis, photoelectrocatalysis, photoelectrostorage and third-generation photovoltaics.

## Methods

### Preparation of the precursor solution

The precursor solution was prepared in a glovebox with O₂ and H₂O level below 5 ppm at room temperature similar to previous reports[19,42]. First, 4 mL of DMSO (99.8%, J&K) was added to a glass vial by a syringe followed by adding 2.4785 g of thiourea (Tu, 99%, Aladdin, recrystallized twice) and 0.7405 g of CuCl (99.999%, Alfa); Second, 1.0420 g of SnCl₄ (99.99%, Alfa) was added to a second vial followed by adding 4 mL of DMSO, 0.6165 g of Zn(OAc)₂ (99.99%, Aladdin), and 0.2638 g of CdCl₂ (99.99%, Aladdin). The two vials were mixed and stirred until clear. The mole ratios of the elements in the solution are Cu/(Zn + Cd +Sn) = 0.85, (Zn + Cd)/Sn = 1.20, Cd/(Zn + Cd) = 0.30.

### Fabrication of the CCZTS absorber film

The glass/Mo/Pd and glass/Mo/Au substrates were prepared by depositing Pd films or Au films on Mo glass substrates using an ion beam sputtering method at room temperature for 15 min. To obtain CCZTS precursor thin film, the precursor solution was spin coated on glass/Mo, glass/Mo/Pd, or glass/Mo/Au substrates at 2000 rpm for 60 s. After spin-coating, the films were immediately annealed at 370 °C for 90 s on a hot plate and then cooled down. The coating and annealing were repeated seven times in order to obtain a desired absorber layer thickness in the glovebox. The films were placed in a graphite box with 100 mg of sulfur powder and 100 mg of stannous sulfide and the graphite box was placed in a rapid thermal processing (RTP) tube furnace for sulfurization. The tube was pumped and refilled

with Argon three times before heated up. The furnace was heated to 620 °C from room temperature over a period of 20 min. The sulfurization was performed at this temperature for 15 minutes and cooled to room temperature naturally.

## Coating of a CdS buffer layer and a Pt co-catalyst on CCZTS films

A CdS buffer layer was deposited on the CCZTS surface by a chemical bath deposition method. Specifically, the CCZTS samples were immersed in aqueous solution containing 1.5 mM $CdSO_4$, 75 mM $SC(NH_2)_2$ and 2.3 M $NH_3 \cdot H_2O$. The deposition temperature was 60 °C and the deposition time was 5 min. Then, a Pt co-catalyst was deposited on the surface of CdS/CCZTS using an ion beam sputtering method at room temperature for 25 s.

## Characterization of samples

The crystalline structures of the three samples were characterized by X-ray diffraction (XRD smartlab, 9 kW). After CCZTS were scraped off the substrate with a tweezer, the $MoS_2$, $MoS_2$/Pd and $MoS_2$/Au substrates were characterized by Grazing Incidence X-Ray Diffraction (GIXRD) (Rigaku Smart lab X-ray diffractometer), four-point probe (Summit 11000 m) and scanning electron microscope (SEM, Gemini500). The cross-section and top-view morphologies of the $MoS_2$/CCZTS, $MoS_2$/Pd/CCZTS and $MoS_2$/Au/CCZTS samples were observed by SEM. The element distribution of the thick and dense CCZTS samples before and after illumination in the electrolyte were identified by time-of-flight secondary ion mass spectrometry (TOF-SIMS 5 iontof, PHI NanoTOFII). The absorption spectra of the samples were recorded by a UV-visible-NIR spectrophotometer (PE lambda 950). The Mott-Schottky plots of CCZTS and $MoS_2$ samples were measured in a cell with a carbon rod and a saturated Ag/AgCl electrode as the counter and reference electrode, respectively. An electrochemical analyzer (CHI 760e, Shanghai Chenhua) was used to control the potentials of CCZTS and $MoS_2$ samples with an a.c. amplitude of 5 mV and a frequency of 1000 Hz. For ultraviolet photoelectron spectra (UPS) measurement, the Pd and Au films were deposited on glass substrates by an ion beam sputtering method for 15 min and the photoelectron spectra of the films were performed in a Kratos Ultra Spectrometer (K-ALPHA+, Thermo Fisher) by using a He I (21.22 eV) discharge lamp.

## In situ irradiated XPS, in situ electrochemical Raman and quasi in situ irradiated Raman characterization of samples

In situ irradiated X-ray photoelectron spectra were measured under a full arc Xe lamp with a XPS spectrometer (Thermofisher Escalab 250Xi). Since the CCZTS layer can attenuate XPS intensity of $MoS_2$, a thin and porous CCZTS were prepared by coating and annealing for two times. For in situ irradiated XPS measurement, the $MoS_2$/CCZTS, $MoS_2$/Pd/CCZTS and $MoS_2$/Au/CCZTS samples were pretreated by immersing them in 0.2 M $Na_2HPO_4$/$NaH_2PO_4$ aqueous solution and exposing them to water vapor. In order to investigate the charge transfer in the three samples, in situ XPS was measured in the dark and after 10-mins' illumination. In situ electrochemical Raman spectra (Horiba T64000, excitation wavelength ~488 nm) were measured in a cell with a saturated Ag/AgCl electrode and a Pt wire as a reference and a counter electrode, respectively. The potential of the $MoS_2$, $MoS_2$/Pd and $MoS_2$/Au was applied by an electrochemical analyzer (CHI-760E, Shanghai Chenhua). Raman signals were detected simultaneously when the electrochemical curves of the samples were measured in the dark. An aqueous solution of 0.2 M $Na_2HPO_4$/$NaH_2PO_4$ ($K_2HPO_4$/$KH_2PO_4$ or $Cs_2HPO_4$/$CsH_2PO_4$) was used as electrolyte. Quasi in situ irradiated Raman spectra of thick and dense CCZTS samples on different substrates were measured before and after illumination by a Xe lamp in 0.2 M $Na_2HPO_4$/$NaH_2PO_4$ aqueous solution.

## Photoelectrochemical measurement

The photoelectrochemical performance was measured in a conventional three-electrode cell using an electrochemical analyzer (CHI-760E, Shanghai Chenhua) with an AM 1.5 G sunlight simulator (CEL-AAAS50, 100 mW/cm², China Education Au-light). A carbon rod and a saturated Ag/AgCl electrode were employed as the counter and the reference electrode, respectively. The electrolyte was the aqueous solution of 0.2 M $Na_2HPO_4$/$NaH_2PO_4$ ($K_2HPO_4$/$KH_2PO_4$ and $Cs_2HPO_4$/$CsH_2PO_4$). The potentials of working electrode were converted to reversible hydrogen electrode by the following formula: $V_{RHE} = V_{Ag/AgCl} + 0.199\ V + 0.059 * pH$. The incident photon-to-current efficiency (IPCE) was obtained under irradiation of different wavelengths of light and was calculated by the following formula:

$$IPCE = \frac{1240 * I_{ph}}{P * \lambda}$$

Where the $I_{ph}$ is the photocurrent density obtained at the potential of 0 $V_{RHE}$, P and $\lambda$ are the power density ($\mu W/cm^2$) and wavelength ($\mu W/cm^2$) of the incident light, respectively.

## Data availability

All data are available in the main text or the Supplementary Information files. Additional data related to the findings of this study are available from the corresponding author upon reasonable request.

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

## Acknowledgements

This work was supported by the National Natural Science Foundation of China (22279052, W. L.), the National Key R&D Program of China (2017YFE0120700, W. L. and 2019YFE0118100, H. X.), the Science and Technology Development Fund from Macau SAR (FDCT 008/2017/AMJ, S. H.-K.).

## Author contributions

W.L. supervised the project, proposed the concept and designed the experiments. H.D. and X.P. carried out sample preparation. H.D. carried out sample characterization and electrochemical measurement. W.L. and H.D. analyzed the data and wrote the paper. Y.G., M.X., P.W., S.H.-K., Y.Y., H.X., W.L. and Z.Z. discussed the results and gave comments on the manuscript.

## Competing interests

The authors declare no competing interests.
