## [Peer Review File · Nature Communications]

REVIEWER COMMENTS

Reviewer #1 (Remarks to the Author):

The manuscript investigates the charge transfer mechanism through semiconductor heterojunctions, using examples of MoS₂/CCZTS systems with Pd or Au interlayers. Band alignment between layers were first considered but the authors argued that it is not consistent with the photocurrent-voltage observation. Na intercalation to MoS₂ was found in systems without interlayer and with Pd interlayer, but it was not observed for the one with Au interlayer. This was correlated with the much higher photocurrent for the system with Au interlayer. The authors explained this phenomena from the suitability of the faradaic potential window within the system that drives interface charge transfer in the favorable direction. The authors proposed this as a general guideline, however, while it is potentially interesting, it is difficult to determine whether the mechanism is indeed applicable to other systems since only specific MoS₂/CCZTS samples were investigated here. In addition, there are a number of issues that are not so clear from the manuscript that the authors should clarify:

1. Line 47-52: the authors referred to the previously reported faradaic junction model. From the formulation of the sentences, it's not clear to me whether the model that they propose here is exactly the same as what's reported or if there's a modification.
2. Line 69: the authors wrote that the potential windows of Au and Pd were measured; this term is not yet introduced at this point. It would be really helpful for the readers (not familiar with faradaic junction model) if this term is explicitly defined.
3. Fig. 1a-c: it is rather difficult to distinguish the layers from the micrographs (especially at the interface). Maybe the authors should label (or partial false-color) the Mo, MoS₂, Pd/Au for better visibility?
4. Supplementary Fig. 9: The linear extrapolation (for flatband and bandgap) is rather subjective (i.e., one can argue that the authors are not taking the correct slope). Perhaps it would be best to provide an error estimate from the various possible extrapolation that one can do with the Mott-Schottky and Tauc plots.
5. Line 151: the authors wrote that the Fermi levels of Pd and Au follow previous study. Did I understand correctly that the values in the reference were taken? If so, given the particulate nature of their Pd and Au layers, the standard reference values may not be valid.

6. Fig. 3: XPS peaks of Mo shift with illumination, which was attributed to photo-reduction towards lower oxidation states. Can't this shift be a band bending effect?

7. The authors used various "in situ" term for their experiments: e.g., in situ XPS, quasi in situ Raman. However, it's not clear what is meant by "in situ" (e.g., directly with electrolyte? environment? applied bias?). If I understood the experimental section correctly, it might be quite a stretch to call the experiments in situ or even quasi in situ. It seems that XPS experiments were still done under standard XPS and Raman environment; except that the samples were pre-treated and/or an illumination source were introduced. But maybe I misunderstood; please therefore clarify.

8. Similarly, the ToF SIMS experiment configuration is not so clear. Did the authors exposed the samples to electrolyte and then brought them out to do the measurements?

9. It is also not clear whether the Na⁺ also intercalates through the dense and thick CCZTS films. If I understood correctly, the XPS and ToF-SIMS experiments were done with the porous and thin CCZTS. Could it be that the mechanism that they proposed (Na⁺ intercalating to MoS₂ to form Na_xMoS₂) only happens with the porous and thin CCZTS? Unless they're suggesting that Na⁺ really intercalates through the CCZTS lattice for the samples with dense CCZTS film.

10. Somewhat related to comment #9 above, Fig. 1a-c seems to suggest that the Au layer is much more dense and close than the Pd (if I guess correctly which one the layer is - see comment #3). Could it be that this morphological feature is simply the reason why it blocks the Na⁺ from intercalating to the MoS₂, and not necessarily the faradaic potential window argument?

11. Also related to comment #9 above, it may be useful for the clarity and to avoid confusion from the readers to explicitly describe whether samples with thin/porous CCZTS or thick/dense CCZTS films were used for the various measurements.

Reviewer #2 (Remarks to the Author):

The present study addresses the study of Au and Pd intermediate layers in MoS₂/CCZTS photocathodes for HER. The study is interesting and targets an interesting concept, which is validating a new theory to rationalize the behaviour of these complex heterostructured photoelectrodes, taking into account ion

intercalation effects, and not only the positions of the energy levels. However, the study should be presented in a more clear language, since it is really difficult for the reader to follow the arguments of the authors in several parts of the manuscript. For this reason, my recommendation is to reject the manuscript and to submit it again, after a careful English revision. More detailed comments are provided below:

1) The terms “semiconductor” and “quantum dot” are not equivalent. Consequently, the authors should rephrase the first paragraph of the introduction to avoid confusion.

2) Supplementary Table 1. The first entry has a “k”, which has not been explained/described. Is this a typo?

3) The quality of the Mott-Schottky plots showed in Supplementary Figure 9 are of poor quality. Furthermore, the manuscript or Supplementary Information do not explain how these measurements are carried out, which is key for the soundness of the manuscript.

4) The authors also claim that the Fermi levels of both Au and Pd are extracted from Ref 26, but in this reference, only workfunctions are reported for these metals. In any case, literature values may not be appropriate for the materials developed in the present study and it is mandatory to obtain reliable experimental data to confirm these values. This is particularly crucial in the present study where the authors try to validate a theory for charge transfer.

5) The explanation of Figure 3 on Raman analysis is particularly difficult to understand. In particular the expression “Au not Pd”

Response to Reviewers' Comments

Firstly, thank all the reviewers for their helpful comments and suggestions.

Reviewer #1:

The manuscript investigates the charge transfer mechanism through semiconductor heterojunctions, using examples of MoS₂/CCZTS systems with Pd or Au interlayers. Band alignment between layers were first considered but the authors argued that it is not consistent with the photocurrent-voltage observation. Na intercalation to MoS₂ was found in systems without interlayer and with Pd interlayer, but it was not observed for the one with Au interlayer. This was correlated with the much higher photocurrent for the system with Au interlayer. The authors explained these phenomena from the suitability of the faradaic potential window within the system that drives interface charge transfer in the favorable direction. The authors proposed this as a general guideline, however, while it is potentially interesting, it is difficult to determine whether the mechanism is indeed applicable to other systems since only specific MoS₂/CCZTS samples were investigated here. In addition, there are a number of issues that are not so clear from the manuscript that the authors should clarify:

Reply: Thank the reviewer for evaluating our study as a potentially interesting work. The reviewer concerns the universality of the mechanism that the suitable potential window can drive interface charge transfer in the favorable direction. In this study, CCZTS and MoS₂ in the MoS₂/CCZTS are a light absorber and an electron acceptor, respectively. In our previous studies (Nat. Commun. 2021, 12, 6363; Angew. Chem. Int. Ed. 2021, 60, 1390-1395; Nat. Commun. 2022, 13, 2544), similar coupled electron and ion transfer was observed in TiO₂/CdS, Si/WO₃, Si/CoO_x heterojunctions in electrolytes (Figure R1). Among these heterojunctions, CdS and Si are light absorbers, TiO₂ ($\text{Ti}^{+4}\text{O}_{2-x}(\text{OH})_{2x} + \text{H}^+ + \text{e}^- \leftrightarrow \text{Ti}^{+3}\text{O}_{1-x}(\text{OH})_{2x+1}$), WO₃ ($\text{WO}_3 + x\text{H}^+ + x\text{e}^- \leftrightarrow \text{H}_x\text{WO}_3$) and CoO_x ($\text{Co}(\text{OH})_2 + \text{h}^+ + \text{OH}^- \leftrightarrow \text{CoOOH} + \text{H}_2\text{O}$) are electron or hole acceptors, respectively. The faradaic potential windows of electron or hole acceptors were also used to describe the interface charge transfer in these heterojunctions. Therefore, the mechanism is universal and applicable to other heterojunctions.

Figure R1. Diagrams for interface charge transfer and faradaic potential windows in TiO_2/CdS (a), Si/WO_3 (b), Si/CoO_x (c) heterojunctions in electrolytes under illumination.

1. Line 47-52: the authors referred to the previously reported faradaic junction model. From the formulation of the sentences, it's not clear to me whether the model that they propose here is exactly the same as what's reported or if there's a modification.

Reply 1: In our previous studies (Nat. Commun. 2021, 12, 6363; Nat. Commun. 2022, 13, 2544; Angew. Chem. Int. Ed. 2021, 60, 1390-1395; iScience 2020, 23, 100949), faradaic junction transfer mechanism (coupled electron and ion transfer) was observed in TiO_2/CdS , Si/WO_3 , Si/CoO_x heterojunctions. In this study, the interface charge transfer in a $\text{MoS}_2/\text{CCZTS}$ heterojunction is similar to the previous studies. However, in this study, we focused on introducing another metal layer (Au or Pd) at the interface of $\text{MoS}_2/\text{CCZTS}$ to improve the photoelectrocatalytic performance remarkably. We found that the faradaic reactions also happened on the surfaces of metal layers during interface charge transfer in the heterojunctions under illumination. Accordingly, a potential window alignment strategy was used to improve the photoelectrocatalytic performance of the heterojunctions in this study for the first time. Therefore, these results are different from our previous studies.

2. Line 69: the authors wrote that the potential windows of Au and Pd were measured; this term is not yet introduced at this point. It would be really helpful for the readers (not familiar with faradaic junction model) if this term is explicitly defined.

Reply 2: The potential window is defined as a potential range in which a material exhibits stable and reversible electrochemical faradaic redox reactions in the electrolyte (Chem. Soc. Rev., 2012, 41, 797-828; Chem. Rev. 2018, 118, 9233-9280). Following the reviewer's suggestion, we have added the definition of potential window into the main text in the revised manuscript.

3. Fig. 1a-c: it is rather difficult to distinguish the layers from the micrographs (especially at the interface). Maybe the authors should label (or partial false-color) the Mo, MoS₂, Pd/Au for better visibility?

Reply 3: Following the reviewer's suggestion, we have labeled the Mo, MoS₂, Pd, Au and CCZTS with different colors for better visibility as follows in the revised manuscript.

Figure R2. The cross-section SEM images of MoS₂/CCZTS (a), MoS₂/Pd/CCZTS (b) and MoS₂/Au/CCZTS (c)

4. Supplementary Fig. 9: The linear extrapolation (for flatband and bandgap) is rather subjective (i.e., one can argue that the authors are not taking the correct slope). Perhaps it would be best to provide an error estimate from the various possible extrapolation that one can do with the Mott-Schottky and Tauc plots.

Reply 4: Following the reviewer’s suggestion, we prepared four samples of CCZTS and MoS₂ and measured their Mott-Schottky curves and UV-visible absorption spectra. The results are shown in Figure R3 and R4, respectively. The average values and errors of the flatbands and bandgaps of the four samples are calculated and shown in Table R1. Moreover, the energy band positions of CCZTS and MoS₂ are calculated from the flatbands and band gaps (Table R1), which are in good agreement with some previous reports (Joule 2018, 2, 537-548; ACS Appl. Energy Mater. 2018, 1, 2749-2757). We have renewed the energy band positions of CCZTS and MoS₂ in the main text and added Figure R3-R4 and Table R1 in Supplementary Information.

Figure R3. Mott-Schottky plots of four CCZTS samples (a-d) and four MoS₂ samples (e-h) in 0.2 M Na₂HPO₄/NaH₂PO₄ aqueous solution in the dark. The Mott-Schottky plots were measured in a cell with a carbon rod and a saturated Ag/AgCl electrode as the counter and reference electrode, respectively. An electrochemical analyzer (CHI 760e, Shanghai Chenhua) was used to control the potentials of CCZTS and MoS₂ samples with an a.c. amplitude of 5 mV and a frequency of 1000 Hz.

Figure R4. The Tauc plots of UV-visible absorption spectra of four CCZTS samples (a-d) and four MoS₂ samples (e-h); A, absorption coefficient; h, Planck's constant; v, photon's frequency.

Table R1. The energy band levels of CCZTS and MoS₂.

	E_{CB}/V_{RHE}	E_{VB}/V_{RHE}	E_g/eV
CCZTS	-0.72 ± 0.05	0.71 ± 0.02	1.45 ± 0.03
MoS ₂	-0.45 ± 0.02	0.79 ± 0.05	1.24 ± 0.03

5. Line 151: the authors wrote that the Fermi levels of Pd and Au follow previous study. Did I understand correctly that the values in the reference were taken? If so, given the particulate nature of their Pd and Au layers, the standard reference values may not be valid.

Reply 5: Indeed, the original values of the Fermi levels of Pd and Au were taken from previous studies (Renew. Sustain. Energy Rev. 2016, 58, 1366-1375; J. Appl. Phys. 1977, 48, 4729-4733). Following the reviewer's suggestion, the Fermi levels (E_F) of Pd and Au were further confirmed by ultraviolet photoelectron spectroscopy (UPS) and the results are shown in Figure R5.

Figure R5. Ultraviolet photoelectron spectra of Au and Pd films on glass substrates after Ar^+ sputter-cleaning (3000 V for 15 minutes). The Pd and Au films were deposited on glass substrates by an ion beam sputtering method for 15 minutes.

The work functions of Pd and Au are calculated to be 5.08 eV and 5.03 eV by subtracting the energy difference between the analyser Fermi levels and the binding energies of the secondary electron cutoffs ($E_{\text{cut off}}$) from the excitation energy of He I excitation energy (21.22 eV) (Nature Catalysis 2020, 3, 932-940) and the results are shown in Table R2. The work functions of Pd and Au particulate films in this study are similar to the reference values in previous studies (Renew. Sustain. Energy Rev. 2016, 58, 1366-1375; J. Appl. Phys. 1977, 48, 4729-4733). The Fermi levels in the reversible hydrogen electrode (RHE) scale can be calculated from the work functions in the absolute vacuum scale by the formula of $eV_{\text{RHE}} = E_{\text{vacuum}} - 4.5 \text{ eV}$ (Science, 2015, 350, 944-948; Nat. Commun. 2021, 12, 6363), which are also summarized in Table R2. We have renewed the Fermi levels of Pd and Au in the main text and added Figure R5 and Table R2 in Supplementary Information.

Table R2. The work functions and Fermi levels (E_{F}) of Pd and Au by ultraviolet photoelectron spectroscopy.

	Work function /eV	Fermi level / V_{RHE}
Pd	5.08	0.58
Au	5.03	0.53

6. Fig. 3: XPS peaks of Mo shift with illumination, which was attributed to photo-reduction towards lower oxidation states. Can't this shift be a band bending effect?

Reply 6: Since Pd and Au indicate similar Fermi levels, the energy band bending of MoS₂/Pd/CCZTS is the same as that of MoS₂/Au/CCZTS (Figure R6). However, the XPS peak shift of Mo was only observed in MoS₂/Pd/CCZTS, but not in MoS₂/Au/CCZTS under illumination. Therefore, the XPS peak shift of Mo cannot come from a band bending effect.

Figure R6. Schematic diagram of energy band bending of MoS₂/Pd/CCZTS (a) and MoS₂/Au/CCZTS (b) heterojunction.

7. The authors used various "in situ" term for their experiments: e.g., in situ XPS, quasi in situ Raman. However, it's not clear what is meant by "in situ" (e.g., directly with electrolyte? environment? applied bias?). If I understood the experimental section correctly, it might be quite a stretch to call the experiments in situ or even quasi in situ. It seems that XPS experiments were still done under standard XPS and Raman environment; except that the samples were pre-treated and/or an illumination source were introduced. But maybe I misunderstood; please therefore clarify.

Reply 7: Following the review's suggestion, the measurement details of XPS and Raman are shown in Figure R7-R9, respectively. For in situ XPS measurement, the heterojunction samples were pretreated by immersing them in 0.2 M Na₂HPO₄/NaH₂PO₄ aqueous solution

(Figure R7a) and were then transferred to the chamber for XPS measurement in the dark (Figure R7b) and under illumination (Figure R7c). The term “in situ” here means that the samples with ions in the electrolyte on the surfaces are measured by XPS under illumination. In order to avoid misunderstanding, we have replaced the “in situ XPS” with “in situ irradiated XPS” in the revised manuscript following previous studies (Adv. Mater. 2019, 31, 1802981; Small 2021, 17, 2103447).

Figure R7. Schematic illustration of the experimental details for in situ XPS. The samples were immersed in 0.2 M Na₂HPO₄/NaH₂PO₄ aqueous solution for pretreatment (a); The XPS signals of the samples were collected in the dark (b) and under full arc Xe lamp illumination (c).

In situ Raman was used to investigate the changes of the samples during electrochemical measurements in the dark. A three-electrode cell was used with the MoS₂, MoS₂/Pd and MoS₂/Au samples as working electrodes (WE), a saturated Ag/AgCl electrode and a Pt wire as a reference (RE) and a counter electrode (CE), respectively (Figure R8). The electrolyte was an aqueous solution of 0.2 M Na₂HPO₄/NaH₂PO₄ (K₂HPO₄/KH₂PO₄ or Cs₂HPO₄/CsH₂PO₄). The potentials of the MoS₂, MoS₂/Pd and MoS₂/Au were controlled by an electrochemical analyzer (CHI-760E, Shanghai Chenhua). The term “in situ” here means that Raman signals were detected simultaneously when the electrochemical curves of the samples were measured in the dark. In order to help the readers to understand the experimental details more easily, we have replaced the “in situ Raman” with “in situ electrochemical Raman” in the revised manuscript following previous studies (ACS Nano 2016, 10, 11344-11350; Nature Protocols 2023, 18, 883-901).

Figure R8. Schematic illustration of the experimental setup for in situ Raman during electrochemical measurement in the dark, where WE, RE, and CE represent the working, reference and counter electrodes, respectively.

Since the light source has an effect on the Raman signals, in situ Raman cannot be used to investigate the interface charge transfer in the heterojunctions under illumination. Therefore, quasi in situ Raman was used and the measurement process is detailed as follows. The heterojunction samples were measured firstly in 0.2 M $\text{Na}_2\text{HPO}_4/\text{NaH}_2\text{PO}_4$ aqueous solution in the dark by Raman (Figure R9a) and then were illuminated under a Xe lamp for 10 minutes (Figure R9b), during which the laser in Raman was off. After illumination, Raman characterization was carried out on the samples immediately (Figure R9c). The term “quasi in situ” here means that the samples in the electrolyte are measured by Raman before and after illumination immediately. In order to avoid readers confusing, we have replaced the “quasi in situ Raman” with “quasi in situ irradiated Raman” in the revised manuscript.

We have added Figure R7-R9 and relevant experimental details in Supplementary Information.

Figure R9. Schematic illustration of the experimental setup for quasi in situ Raman in the electrolytes under illumination. The Raman spectra of the samples were collected in 0.2 M $\text{Na}_2\text{HPO}_4/\text{NaH}_2\text{PO}_4$ aqueous solution in the dark (a), and then the samples were exposed to a Xe lamp for 10 minutes (b), the Raman characterization was carried out on the samples immediately after illumination (c).

8. Similarly, the ToF-SIMS experiment configuration is not so clear. Did the authors exposed the samples to electrolyte and then brought them out to do the measurements?

Reply 8: As the reviewer mentioned, the samples were immersed in 0.2 M $\text{Na}_2\text{HPO}_4/\text{NaH}_2\text{PO}_4$ aqueous solution for 20 minutes and then brought out to do the ToF-SIMS measurements. We have added the relevant experimental details in Supplementary Information.

9. It is also not clear whether the Na^+ also intercalates through the dense and thick CCZTS films. If I understood correctly, the XPS and ToF-SIMS experiments were done with the porous and thin CCZTS. Could it be that the mechanism that they proposed (Na^+ intercalating to MoS_2 to form Na_xMoS_2) only happens with the porous and thin CCZTS? Unless they're suggesting that Na^+ really intercalates through the CCZTS lattice for the samples with dense CCZTS film.

Reply 9: The XPS experiments were done with the porous and thin CCZTS in order to obtain signals of CCZTS and MoS_2 at the same time, while the ToF-SIMS experiments were done with thick and compact CCZTS. The results suggest that Na^+ can also intercalate through the dense and thick CCZTS films. In higher magnification SEM images, some pin

holes inevitably exist on the surfaces of the CCZTS films prepared by the spin-coating method (Figure R10). Similar morphologies of pin holes are also observed on the CZTS samples by the same method in previous studies (Energy Environ. Sci., 2021, 14, 2369-2380; Science Bulletin, 2020, 65, 738-746). Therefore, Na^+ can also intercalate into MoS_2 through the pin holes in the dense and thick CCZTS films. We have added the above discussion in the main text and Figure R10 in Supplementary Information in the revised manuscript.

Figure R10. Top-view SEM images of CCZTS on MoS_2 at different scales of 500 nm (a), 1 μm (b) and 2 μm (c).

10. Somewhat related to comment #9 above, Fig. 1a-c seems to suggest that the Au layer is much more dense and close than the Pd (if I guess correctly which one the layer is - see comment #3). Could it be that this morphological feature is simply the reason why it blocks the Na^+ from intercalating to the MoS_2 , and not necessarily the faradaic potential window argument?

Reply 10: As the reviewer mentioned in comment #3, it is difficult to distinguish the morphological features of Au and Pd layers from the cross-section SEM images of $\text{MoS}_2/\text{Pd}/\text{CCZTS}$ and $\text{MoS}_2/\text{Au}/\text{CCZTS}$. However, it is clearly seen that both Pd and Au intermediate layers indicate similar particles in the two heterojunction samples from the top-view SEM images by removing the upper layers of CCZTS (Figure R11). Therefore, the morphological feature is not the reason that the Au intermediate layer blocks the Na^+ from intercalating to the MoS_2 . We have added the above discussion in the main text in the revised manuscript.

Figure R11. Top-view SEM images (a-c) and corresponding EDS characterization (d-f) of MoS₂, MoS₂/Pd and MoS₂/Au, after scrapping off the upper CCZTS from the substrates by a tweezer.

11. Also related to comment #9 above, it may be useful for the clarity and to avoid confusion from the readers to explicitly describe whether samples with thin/porous CCZTS or thick/dense CCZTS films were used for the various measurements.

Reply 11: Following the reviewer's suggestions, we have explicitly described the samples with thin/porous CCZTS or thick/dense CCZTS films that were used for the various measurements in the revised manuscript.

Reviewer #2:

The present study addresses the study of Au and Pd intermediate layers in MoS₂/CCZTS photocathodes for HER. The study is interesting and targets an interesting concept, which is validating a new theory to rationalize the behaviour of these complex heterostructured photoelectrodes, taking into account ion intercalation effects, and not only the positions of the energy levels. However, the study should be presented in a more clear language, since it is really difficult for the reader to follow the arguments of the authors in several parts of the manuscript. For this reason, my recommendation is to reject the manuscript and to

submit is again, after a careful English revision. More detailed comments are provided below:

Reply: Thank the reviewer for evaluating our study as an interesting work. The reviewer mainly concerns the unclear language in the original manuscript. Therefore, we have revised English very carefully to make it easier for readers to understand the content. Moreover, our point-by-point responses to the reviewer's comments are listed as follows.

1. The terms “semiconductor” and “quantum dot” are not equivalent. Consequently, the authors should rephrase the first paragraph of the introduction to avoid confusion.

Reply 1: Following the reviewer's suggestions, we have rephrased the first paragraph of the introduction to avoid confusion in the revised manuscript as follows.

In the past several decades, photocatalysis, photoelectrocatalysis, photoelectrostorage, third-generation photovoltaics (such as quantum-dot or dye-sensitized, organic and perovskite solar cells) have attracted wide interest due to their low cost and high efficiency for converting solar energy into chemical or electric energy¹⁻⁵. Among them, different kinds of heterojunctions, such as type-II heterojunction⁶ and Z-scheme junction⁷, have been widely used to separate photo-excited carriers in light absorbers (Supplementary Fig. 1). In all of these previous studies⁶⁻⁸, a band alignment theory is a basis for manipulating interface electron transfer in high-performance devices. However, Luo et al. have found that faradaic junction interface charge transfer process (coupled electron and ion transfer) widely exists in these solar energy conversion and storage devices⁹⁻¹⁷. Since the classic band alignment theory only regulates electron transfer, it is not suitable for explaining coupled electron and ion transfer in faradaic junctions. Therefore, it is very significant to explore a new design concept for regulating coupled electron and ion transfer in order to improve the performance of the semiconductor heterojunctions.

2. Supplementary Table 1. The first entry has a “k”, which has not been explained/described. Is this a typo?

Reply 2: The “k” is an abbreviation for kilo, which means “thousand” (Adv. Funct. Mater. 2018, 1703369). In order to avoid misunderstanding, we have replaced the symbol of “k” with the number of “1000” in the revised manuscript as follows (Table R3).

Table R3. Sheet resistance of MoS₂, MoS₂/Pd and MoS₂/Au.

	Sheet resistance / $\Omega \cdot \text{sq}^{-1}$
MoS ₂	> 15000
MoS ₂ /Pd	42
MoS ₂ /Au	32

3. The quality of the Mott-Schottky plots showed in Supplementary Figure 9 are of poor quality. Furthermore, the manuscript or Supplementary Information do not explain how these measurements are carried out, which is key for the soundness of the manuscript.

Reply 3: A similar comment has been replied to Question 4 from Reviewer 1. Following the reviewer’s suggestion, we prepared four samples of CCZTS and MoS₂ and measured their Mott-Schottky curves. The results are shown in Figure R12. The average values of the flatbands of the four samples are close to the values in the main text, and the errors are very small (<0.05 V).

Figure R12. Mott-Schottky plots of CCZTS (a) and MoS₂ (b) in 0.2 M Na₂HPO₄/NaH₂PO₄ aqueous solution.

The Mott-Schottky measurement details are described as follows. A three-electrode cell was used to measure the Mott-Schottky curves with a carbon rod and a saturated Ag/AgCl electrode as the counter and reference electrode, respectively. The electrolyte was 0.2 M Na₂HPO₄/NaH₂PO₄ aqueous solution (pH~6.5). The potentials of CCZTS and MoS₂ samples were controlled by an electrochemical analyzer (CHI 760e, Shanghai Chenhua) with an a.c. amplitude of 5 mV and a frequency of 1000 Hz.

We have renewed the energy band positions of CCZTS and MoS₂ in the mechanism of interface charge transfer in the main text and added Figure R12 and relevant experimental details in Supplementary Information.

4. The authors also claim that the Fermi levels of both Au and Pd are extracted from Ref 26, but in this reference, only work functions are reported for these metals. In any case, literature values may not be appropriate for the materials developed in the present study and it is mandatory to obtain reliable experimental data to confirm these values. This is particularly crucial in the present study where the authors try to validate a theory for charge transfer.

Reply 4: A similar comment has been replied to Question 5 from Reviewer 1. As the reviewer mentioned, it is important to obtain reliable experimental data to confirm the

Fermi levels of Au and Pd. Therefore, following the reviewer's suggestion, the Fermi levels (E_F) of Pd and Au were further confirmed by ultraviolet photoelectron spectroscopy (UPS) and the results are shown in Figure R13.

Figure R13. Ultraviolet photoelectron spectra of Au and Pd films on glass substrates after Ar^+ sputter-cleaning (3000 V for 15 minutes). The Pd and Au films were deposited on glass substrates by an ion beam sputtering method for 15 minutes.

The work functions of Pd and Au are calculated to be 5.08 eV and 5.03 eV by subtracting the energy difference between the analyser Fermi levels and the binding energies of the secondary electron cutoffs ($E_{\text{cut off}}$) from the excitation energy of He I excitation energy (21.22 eV) (*Nat. Catal.* 2020, **3**, 932-940) and the results are shown in Table R4. The work functions of Pd and Au particulate films in this study are similar to the reference values in previous studies (*Renew. Sustain. Energy Rev.* 2016, **58**, 1366-1375; *J. Appl. Phys.* 1977, **48**, 4729-4733). The Fermi levels in the reversible hydrogen electrode (RHE) scale can be calculated from the work functions in the absolute vacuum scale by the formula of $eV_{\text{RHE}} = E_{\text{vacuum}} - 4.5 \text{ eV}$ (*Science*, 2015, **350**, 944-948; *Nat. Commun.* 2021, **12**, 6363), which are summarized in Table R4. We have renewed the Fermi levels of Pd and Au in the main text and added Figure R13 and Table R4 in Supplementary Information.

Table R4. The work functions and Fermi levels (E_F) of Pd and Au by ultraviolet photoelectron spectroscopy.

	Work function /eV	Fermi level / V_{RHE}
Pd	5.08	0.58
Au	5.03	0.53

5. The explanation of Figure 3 on Raman analysis is particularly difficult to understand. In particular the expression “Au not Pd”.

Reply 5: Following the reviewer’s suggestions, we have revised the relevant description of Raman analysis as follows.

Quasi in situ irradiated Raman spectra were also used to investigate the effects of Au and Pd on the reduction process of MoS₂ in MoS₂/CCZTS before and after illumination³¹ and the experimental details are shown in Supplementary Fig. 17. The Raman peaks at 382 cm⁻¹ and 408 cm⁻¹ are assigned as E_{12g} and A_{1g} vibrational modes of MoS₂, and the peak at 332 cm⁻¹ corresponds to CCZTS^{22,29}. After MoS₂/CCZTS is illuminated, the Raman peak intensity of CCZTS does not change, while the Raman peak intensity of MoS₂ decreases (Fig. 3d). In contrast, the Raman peak intensity of MoS₂ film alone does not decrease after the same illumination (Supplementary Fig. 18). Therefore, interface charge transfer in the MoS₂/CCZTS heterojunction under illumination leads to the change of the Raman peak intensity of MoS₂. According to previous studies^{32,33}, the decrease of Raman peak intensity of MoS₂ can be attributed to the Na⁺ intercalation into MoS₂. The Raman peak intensity of MoS₂ decreases in MoS₂/Pd/CCZTS but not in MoS₂/Au/CCZTS after illumination (Figs. 3e-f). The results further suggest that the Au rather than Pd intermediate layer can inhibit the reduction intercalation reaction of MoS₂ in the MoS₂/CCZTS junction, which are in good agreement with in situ irradiated XPS results mentioned above.

REVIEWERS' COMMENTS

Reviewer #1 (Remarks to the Author):

The authors have addressed all my comments on the initial manuscript. I have no remaining objection on the publication of the revised manuscript.